# Proteomic Profiles of the Longissimus Muscles of Entire Male and Castrated Pigs as Related to Meat Quality

**DOI:** 10.3390/ani9030074

**Published:** 2019-02-27

**Authors:** Martin Škrlep, Urška Tomažin, Nina Batorek Lukač, Klavdija Poklukar, Marjeta Čandek-Potokar

**Affiliations:** 1Agricultural Institute of Slovenia, Hacquetova ulica 17, 1000 Ljubljana, Slovenia; martin.skrlep@kis.si (M.Š.); urska.tomazin@kis.si (U.T.); nina.batorek@kis.si (N.B.L.); klavdija.poklukar@kis.si (K.P.); 2University of Maribor, Faculty of Agriculture and Life Sciences, Pivola 10, 2311 Hoče, Slovenia

**Keywords:** pigs, entire males, castrates, meat quality, proteomic profile

## Abstract

**Simple Summary:**

The meat (loin muscle) of entire male pigs and barrows (surgical castrates) was analyzed for various properties to understand the etiology of differences in quality. Proteomic analysis indicated a higher level of proteolysis in the entire male pigs. Nevertheless, their meat exhibited more toughness, which could be associated with lower intramuscular fat and lower water holding capacity, the latter resulting from higher levels of protein oxidation.

**Abstract:**

There are indications of reduced meat quality in entire male pigs (EMs) in comparison to surgically castrated pigs (SCs); however, the differences are not strongly confirmed, and the etiology is not clarified. In the present study, samples of the longissimus dorsi, pars lumborum muscle (LL) from EMs (*n* = 12) and SCs (*n* = 12) of the same age and weight were evaluated at the physico-chemical and proteomic level. EMs exhibited lower intramuscular fat content, higher collagen content with higher solubility, a higher level of protein carbonyl groups (indicating higher protein oxidation), lower water holding capacity, and tougher meat than SCs. Proteomic analysis revealed differences in heat shock proteins expression, while a greater abundance of several other identified proteins (malate dehydrogenase, Na/K-transporting adenosintriphosphatase (ATP-ase) subunit alpha-1, and blood plasma proteins) indicates that EMs have a more oxidative metabolic profile than that of SCs. More abundant protein fragments (mainly actin and myosin heavy chain) suggest a higher degree of proteolysis in EMs, which was not followed by lower meat toughness.

## 1. Introduction

Surgical castration of male piglets intended for fattening is currently the most common practice in the majority of developed, pig-breeding countries. The main reason for conducting this procedure is the prevention of boar taint, in addition to the reduction of aggressive behavior and the improvement of meat quality [1]. However, due to the strong public initiative to stop surgical castration in the European Union (EU) [2], rearing entire male pigs (EMs) might become the more common practice in Europe. Despite the fact that there are several positive features of this alternative compared with surgical castration, including better feed conversion, higher lean deposition, and, consequently, higher cost-effectiveness [3], there are also some disadvantages. In addition to more aggressive behavior [4] and the increased risk of boar taint [5], the possibility of reduced meat quality presents an important concern. A review by Lundström et al. [5] pointed out the problem of extreme carcass leanness (in relation to soft fat, lack of tissue cohesion, and a high proportion of unsaturated fatty acids) and a higher incidence of dark, firm, and dry (DFD) meat in EMs. In addition, the literature data, summarized in two meta-analytical studies [6,7], showed reduced intramuscular fat content (IMF), increased meat toughness, and lower ultimate pH in EMs than in surgically castrated pigs (SCs), whereas no effect was reported for other meat quality traits. However, several recent studies [8,9,10] have reported that EM meat has inferior water holding capacity, but the results of these studies are not fully consistent regarding meat quality and need further substantiation. Moreover, the etiology of the biochemical processes associated with altered meat quality has not been elucidated. Therefore, in the present study, an approach based on biochemical analyses and two-dimensional (2D) electrophoresis of meat (specifically the longissimus dorsi, pars lumborum muscle (LL muscle)) was used to try to characterize and associate the potential differences between EMs and SCs with respect to meat quality traits in their proteomic profiles.

## 2. Materials and Methods 

This work was undertaken within the normal running of a farm (respecting the Slovenian law on animal protection). No procedures that would demand ethical protocols according to Directive 2010/63/EU (2010) were performed on the pigs used in this study. Moreover, all the tissue (meat) samples were taken after slaughter. 

The material for the present study consisted of 12 pigs of each sex group of the same crossbreed (Landrace × Large White), which were raised in equivalent conditions (i.e., on an intensive commercial farm with a corn-based commercial diet of 16% crude protein and 13.1 MJ ME/kg, ad libitum feeding) and slaughtered at a similar age (198 ± 4 days) in the same abattoir applying routine slaughter procedure (i.e., CO_2_ stunning, vertical exsanguination, vapor scalding, and evisceration). 

At the end of the slaughter line, the carcasses were weighed and classified according to the method approved in Slovenia [11], which uses the measurements of back fat (minimal thickness of the fat on the top of the gluteus medius muscle) and muscle thickness (the shortest distance between the dorsal edge of the vertebral canal and the cranial end of the gluteus medius) for the estimation of lean meat percentage. After being chilled for 24 h, the carcasses were cut at the last rib, and the samples of the longissimus dorsi, pars lumborum muscle (LL) were taken for physico-chemical and proteomic analyses. The objective color (CIE (International Commission on Illumination) L*, a*, and b* parameters), ultimate pH, thawing loss, cooking loss, and shear force (WBSF) were assessed as described in the work of Batorek et al. [8]. The objective color parameters were measured on the freshly-cut LL surface using a Minolata CR-300 (Minolta Co. Ltd., Osaka, Japan). A measurement of pH was performed using a MP120 Mettler-Toledo pH meter (Mettler-Toledo GmbH, Schwarzenbach, Switzerland). For drip loss (assessed according to the EZ-DripLoss method [12]), 2 cylindrical samples of approximately 10 g were cut from the center of the LL and stored in plastic containers for 24 h at 4 °C. The drip loss was calculated as the difference between the initial sample weight and the weight after storage. For the determination of thawing loss, cooking loss, and shear force, a 2.5 cm thick chop of the LL was vacuum packed and stored frozen at −20 °C. The thawing loss was determined from the difference in weight after thawing (overnight at 4 °C). The same sample was afterwards cooked to the internal temperature of 72 °C using a thermostatic bath (ONE 7-45, Memmert GmbH, Schwabach, Germany), reweighed for the determination of the cooking loss, cooled overnight (4 °C), and used for WBSF determination. Then, two cylindrical cores (2.5 cm thick) were excised from the central part of the LL, and the shear force was measured using a TA Plus texture analyzer (Ametek Lloyd Instruments Ltd., Bognor Regis, UK). The moisture, protein, and intramuscular fat (IMF) content were determined by near-infrared spectroscopy (NIR Systems 6500, Foss NIR System, Silver Spring, MD, USA), applying internal calibrations developed at the Agricultural Institute of Slovenia. Samples of the LL for proteomic analyses, protein oxidation, and collagen and myoglobin content analyses were frozen in liquid nitrogen and stored at −80 °C until analyzed. Protein oxidation (i.e., protein carbonyl group content) was determined according to the method used by Traore et al. [13]. Shortly, after myofibrillar isolation, the samples were treated with 2,4-dinitrophenylhydrazine (DNPH) dissolved in hydrochloric acid. The proteins were precipitated by trichloroacetic acid, washed with ethanol and ethyl acetate to eliminate all the residual DNPH, and dissolved in guanidine hydrochloride solution in a phosphate buffer. The protein carbonyl group content was calculated from the absorbance measured at 370 nm (using BioSpectrometer Fluorescence, Eppendorf GmbH, Wesseling-Berzdorf, Germany). The myoglobin concentration was analyzed according to the method of Trout [14]. The muscle samples were homogenized in a potassium phosphate buffer, filtered, and added to Triton X-100 and sodium nitrite solution. The myoglobin concentration was calculated from the absorbance measured at 370 nm and 409 nm (with BioSpectrometer Fluorescence, Eppendorf GmbH, Wesseling-Berzdorf, Germany). The total collagen was determined as hydroxyproline content multiplied by a factor of 8 (according to ISO (International Organization for Standardization) 3496 [15]). Briefly, samples of the muscle (previously cooked at 77 °C for 90 min in 25% Ringer’s solution) were incubated with sulfuric acid at 105 °C overnight. The resulting hydrolysate was filtered and incubated with chloramine-T and p-dimethylaminobenzaldehyde in perchloric acid and propan-2-ol. The hydroxyproline content was determined from the absorbance measured at 558 nm (with BioSpectrometer Fluorescence, Eppendorf GmbH, Wesseling-Berzdorf, Germany). For the insoluble collagen fraction, muscle samples were heated in Ringer’s solution (90 min at 77 °C) and centrifuged (4000× *g*, 10 min, 20 °C), and the supernatant was discarded. The pellet was further analyzed as described for total collagen. The soluble collagen content was calculated from the difference between the total and insoluble collagen. The collagen solubility was calculated as a ratio between soluble and total collagen.

The proteomic analysis consisted of 2-dimmensional electrophoresis and gel image analysis. Prior to proteomic analysis, the muscle samples were powdered in liquid nitrogen and cleaned of impurities using a 2D Clean-UP-Kit (GE Healthcare Bio-Sciences AB, Uppsala, Sweden) according to the manufacturer’s instructions. The obtained proteins were then dissolved in an extraction buffer (7 M urea, 2 M thiourea, 4% CHAPS (3-[(3-Cholamidopropyl) dimethylammonio]-1-propanesulfonate hydrate) (w/v), and 1% DTT (dithiothreitol) (w/v)) and stored at −80 °C until use. The protein concentration of the samples was determined by Bradford protein assay (Bio-Rad, CA, USA) after diluting the samples 50:1 in water. Prior to the isoelectric focusing, 800 μg of the protein samples were diluted in rehydration buffer (7 M urea, 2 M thiourea, 0.5% carrier ampholytes (v/v), 2% CHAPS (w/v), 1% DTT (w/v), and bromphenol blue), loaded on ImmobilineTM DryStrips (GE Healthcare Bio-Sciences AB, Uppsala, Sweden; pH 3-11 non-linear, 24 cm) using a dry strip reswelling tray, and left to rehydrate for 16 h. The rest of the procedure was identical to that described previously in [16]. Shortly thereafter, the rehydrated strips were submitted for isoelectric focusing (using an Ettan IGPhor 3 IEF, GE Healthcare Bio-Sciences AB, Uppsala, Sweden) and afterwards equilibrated in two different buffers containing DTT and iodoacetamide. For the separation of the proteins according to their molecular weight, SDS-PAGE was performed using Ettan Dalt Six unit (GE Healthcare Bio-Sciences AB, Uppsala, Sweden) on 12.5 % polyacrylamide gels, and the proteins were stained with Coomassie Brilliant Blue G250. For each sample, two technical repetitions were made, resulting in 48 gels. The gel images were digitalized using an Image Scanner III (GE Healthcare Bio-Sciences AB, Uppsala, Sweden) and analyzed by the ImageMaster 2D Platinum (version 6) computer program (GE Healthcare Bio-Sciences AB, Uppsala, Sweden). After detection, spots were automatically matched to the master gel. In addition, a manual control of the process was performed, assuring the high quality of the matching and the creation of an accurate master gel. To minimize the differences due to protein loading and staining, relative spot volumes were calculated, representing the ratio between the individual spot volume and summarized volume of all the spots on the gel. Overexpressed and undefinable spots were excluded from the analysis. In addition, only spots present in both technical and all the biological repetitions (i.e., animals) were considered. The average of both technical repetitions was calculated, and the data were log-transformed and used in statistical analysis.

For the identification, spots of interest were excised from the gel and analyzed by mass spectrometry (MALDI-TOF-TOF) at York Technology Facility (University of York, UK). The spots were washed twice in a 50% (v/v) aqueous acetonitrile solution containing 25 mM ammonium bicarbonate solution, which was followed by dehydration in 100% acetonitrile and drying under vacuum for 20 min. For in-gel tryptic digestion, proteomic grade trypsin was dissolved in 50 mM acetic acid and diluted 5-fold with 25 mM ammonium bicarbonate resulting in a 0.02 μg/μL trypsin concentration. The spots were rehydrated in 10 μL of trypsin solution for 10 min when an adequate volume of 25 mM ammonium bicarbonate solution was added to cover the spots and left to hydrolyze overnight at 37 °C. The resulting peptide extract (1 μL) mixture was applied to a ground steel MALDI target plate and immediately followed by an equal volume of a freshly prepared solution of 4-hydroxy-α-cyano-cinnamic acid solution (10 mg/mL) in 50% (v/v) aqueous acetonitrile solution containing 0.1% trifluoroacetic acid (v/v). Positive-ion MALDI mass spectra were obtained on a Bruker ultraflex III (Bruker Daltonics Ltd., Coventry, UK) in reflectron mode, equipped with a Nd/YAG smart beam laser and acquired over a range of 800–5000 m/z. An external calibration of the final mass spectra was performed against an adjacent spot containing 6 peptides: des-Arg1-Bradykinin (m/z 904.681), Angiotensin I (m/z 1296.685), Glu1-Fibrinopeptide B (m/z 1750.677), ACTH (1-17 clip, m/z 2093.086), ACTH (18-39 clip, m/z 2465.198), ACTH (7-38 clip, m/z 3657.929). The monoisotopic masses were obtained using a SNAP (smart numerical annotation procedure) averaging algorithm (C 4.9384, N 1.3577, O 1.4773, S 0.0417, and H 7.7583) and a S/N threshold of 2. For each spot, the ten strongest precursors with a S/N greater than 30 were selected for MS/MS fragmentation, which was performed in LIFT (ion potential energy raising) mode without the introduction of a collision gas. The default calibration was used for MS/MS spectra, which were baseline-subtracted and smoothed (Savitsky-Golay, width 0.15 m/z, 4 cycles); the monoisotopic peak detection used a SNAP averaging algorithm (C 4.9384, N 1.3577, O 1.4773, S 0.0417, and H 7.7583) with a minimum S/N of 6. The processing of the spectra and the generation of the peak list were performed using Bruker flexAnalysis software version 3.3. The tandem mass spectral data were submitted to a database search using a locally running copy of the Mascot program (version 2.4; Matrix Science Ltd., London, UK) and the Bruker ProteinScape interface (version 2.1). The criteria specified to search the database (UniProt_Pig_SP, version 20141110, 1413 seq, 521784 res) allowed one missed trypsin cleavage, cystein carbamidometylation (set to fixed modification), methionine oxidation and asparagine and glutamine deamination (set to variable modifications), peptide tolerance ± 100 ppm, and MS/MS tolerance ± 0.5 Da. The results were filtered to accept only peptides with an expected score of 0.05 or lower.

The statistical analysis of the data was performed using the general linear models (GLM) procedure of SAS statistical software (SAS institute Inc., Cary, NC, USA), including a fixed effect of sex (i.e., entire male vs surgical castrate). In the case of a significant (*p* < 0.05) effect, the differences between the EMs’ and SCs’ least square means were compared using the PDIFF option. A trend of the effect was considered to occur when *p* < 0.10.

## 3. Results

### 3.1. Carcass and Meat Physico-Chemical Traits

Despite having similar weights at slaughter, EMs exhibited lower (*p* < 0.05) carcass weight, lower backfat, and lower muscle thickness than SCs (Table 1). 

Concerning meat chemical composition (Table 2), EMs had lower (*p* < 0.05) IMF and higher (*p* < 0.05) moisture content than did SCs. There were also notable differences (*p* < 0.05) between the two sex groups with respect to collagen content and solubility with 2.2-fold higher values for the collagen content and 2-fold more soluble collagen observed in EMs than SCs. In addition, EMs exhibited 2-fold higher levels (*p* < 0.05) of carbonyls, i.e., protein oxidation, than SCs. 

There were also several differences in meat quality traits (Table 3). Drip loss, cooking loss, and meat toughness (assessed as shear force) were significantly higher (*p* < 0.05) in EMs than in the SC samples. EMs also exhibited lighter (L*) and more yellow (b*) meat than SCs (*p* < 0.05), with a trend (*p* < 0.10) towards lower redness (a*).

### 3.2. Proteomic Profile

More than 1000 individual protein spots were detected on the gels, but due to the missing values, low quality, or excessive saturation, only 442 spots were included in the statistical analysis. A total of 124 spots (Figure 1) were differentially expressed (*p* < 0.05) between EMs and SCs, with the majority of the spots (*n* = 104) having higher abundance in EMs than SCs. 

Among those, there were 30 spots with 1.5-fold or higher abundance in EMs than in SCs. With regard to the spots that were more abundant in SCs, there were 20 with four of them being 1.5-fold or more abundant in SCs than in EMs. From this pool of spots, the spots of interest (chosen taking into account the difference in relative abundance between EMs and SCs and with sufficient spot intensity) were excised, and 32 of them were identified by mass spectrometry. Among the identified spots (Table 4 and Figure 2), there were 13 myofibrillar proteins, 10 blood plasma proteins, six metabolic enzymes, and three chaperone regulatory proteins. According to the differences between the theoretical and estimated molecular weights, there were 18 spots that could be undeniably asserted as protein fragments. The rest of the spots, where the observed difference was small, were considered to be entire protein molecules. Among the spots (*n* = 26) that exhibited higher abundance in EMs than in SCs, there were 11 entire proteins; five of them were identified as serum albumin and the other six as slow skeletal muscle troponin T, fast skeletal muscle troponin T, serotransferrin, heat shock protein 70 kDa, malate dehydrogenase, and Na/K-transporting adenosintriphosphatase (ATP-ase) subunit alpha-1. The remaining 15 spots were identified as fragments of skeletal muscle α-actin (*n* = 6), myosin heavy chain 2a (*n* = 3), creatine kinase (*n* = 2), coagulation factor VIII (*n* = 2), beta enolase (*n* = 1), and Na/K-transporting ATP-ase, subunit alpha-1 (*n* = 1). As for the spots (*n* = 6) showing higher abundance in SCs than in EMs, there were three entire proteins, two of them identified as α-crystallin B and one as serum albumin, while the remaining three spots were recognized as fragments (two of them as skeletal muscle α-actin and one as coagulation factor VIII).

## 4. Discussion

Considering that the live weight in EM and SC pigs was not different, the observed lower EM carcass weight can be at least partly attributed to the bigger reproductive tract in EMs (testes and accessory glands), which is in agreement with several studies showing the lower killing out percentage of EMs than SCs [17,18,19]. The present study confirms that there is lower fat deposition in EMs than in SCs [6,7]. 

Our results indicate more developed connective tissue in EMs. There are few studies showing the differences in collagen content between pig sexes, whereas in cattle, bulls are known to have higher collagen content than (castrated) steers [20,21,22]. A higher amount of collagen in the longissimus dorsi muscle of EMs than that in SC or gilts was reported [23,24], and this was related to the male hormone testosterone [24], which is in agreement with the present study. In line with these observations, there is also evidence of stronger dermis development, thicker skin [25], and a higher amount of collagen in the backfat of EMs [26]. Though EMs have more collagen, it is more soluble, denoting more immature collagen with less cross-linking, which may be related to the higher protein turnover reported in young EMs [27]. 

The markedly higher concentration of protein carbonyl groups in EMs than in SCs demonstrates that proteins were more oxidized in EMs than in SCs. In the present study, the fatty acid profile of fat tissue was not assessed; however, it can be expected that EMs have more unsaturated fat than SCs due to their lower backfat thickness and IMF [28]. Protein carbonyl group concentration is positively correlated with fat oxidation, as the oxidation of lipids (especially unsaturated ones) is one of the main factors governing the oxidation of proteins and amino acids [29]. The higher objective color parameter b* (i.e., yellowness) and lower a* (redness) values observed in EM muscle than in SC muscle are also indicative of higher oxidation levels [30,31,32]. 

There was a trend of lower values of a* in EMs that might also be related to differences in myoglobin content [30]; however, the latter was not significant (only numerically lower in EMs compared with SCs). This agrees with the majority of studies investigating EM meat, which reported no [7,9,10,33] or very small differences [18] in color parameters between EMs and SCs. There are only two studies showing that EMs exhibit darker [8] or redder meat [34], which might be also due to pH value differences. Additionally, a higher proportion of DFD meat has been indicated in the case of EMs [5].

The majority of the studies showed no differences between EMs and SCs; however, several recent studies have pointed out the significantly reduced water retention ability of meat from EMs, either measured as drip [8,9,10] or cooking loss [8,10]. Inferior water holding capacity can be related to increased protein oxidation. Oxidation can lead to changes in the physical properties of muscular proteins, including loss of solubility, aggregation, denaturation, cross-linking, and myofibril shrinkage, consequently lowering the ability of muscular structures to bind or hold water [29]. Reduced water holding capacity and protein oxidation have been held responsible for increased meat toughness [35], which corroborate the results of the present study. The greater toughness of EM meat may be related to the significant correlation between shear force and IMF (r = −0.57, *p* = 0.003) and between shear force and carbonyl groups content (r = 0.51, *p* = 0.010), whereas no significant correlations with shear force could be observed for either total collagen content (r = −0.07, *p* = 0.752) or collagen solubility (r = −0.21, *p* = 0.325). Moreover, within the EM group, a significant negative correlation (r = −0.69, *p* = 0.013) between shear force and collagen content was observed, denoting that collagen content cannot explain the increased meat toughness in EMs.

Proteomic analysis allowed the identification (by mass spectrometry) of a limited number of protein spots and thus of differentially expressed proteins between EMs and SCs. Still some interesting observations and conclusions could be drawn. There was a higher abundance of protein fragments in EMs than in SCs, which is indicative of a higher level of either in vivo or post mortem proteolysis in EMs. Previous proteomic studies on pig muscle also detected myofibrillar protein fragments (similar to that of actin and myosin) [36,37,38,39]. Sarcoplasmic protein fragments similar to those of enolase and muscle creatine kinase were also reported for aged muscles [36,40,41]. Sarcoplasmic proteins were (besides myofibrillar) identified as one of the possible targets for the calpain proteolytic system [42,43,44] that is commonly believed to have a major role in post mortem meat tenderization [45]. Generally, muscle proteolytic potential (i.e., proteolytic enzymes activities) is positively correlated with protein turnover and the level of protein deposition [46]. Due to steroid hormones, protein anabolic potential is notably higher in EMs than in SCs [27], which could explain the 5-fold higher incidence of identified protein fragments (i.e., 15 vs 3; Table 4) in EMs than in SCs in the present study. However, to the best of our knowledge, so far, no literature has clearly related EMs with increased activity of proteolytic enzymes. In our recent research analyzing dry-cured hams from EMs [47], EMs with higher androsterone in fat tissue had a higher proteolysis index (% of non-protein nitrogen), which is indicative of the association between androgens and proteolysis. The higher abundance of protein fragments in EMs than in SCs in the present study did not result in the higher tenderness of EM meat. Although several proteomic studies on porcine [37,39,48] and bovine muscles [49,50,51] showed that actin and myosin proteins may be degraded, it was concluded that they are not broken down to any greater extent during post mortem storage, and this may not be the primary cause for meat tenderization [39,52]. Perhaps a short post mortem storage time (24 h in the present study) was not enough to detect the differences in proteolysis. In addition, it is possible that other factors, such as protein oxidation, could have interfered, as shown for myosin heavy chain, which may form molecular cross-links under oxidizing conditions thus increasing shear force and decreasing protein solubility and possibly water holding capacity [52].

With regard to the identified entire protein molecules, the one showing the highest difference (1.9-fold more expressed in EMs than in SCs) was the 70 kDa heat shock protein (HSP70). Other identified heat shock proteins (two spots identified as α-crystallin B, a protein from the small heat shock protein (sHSP) family) were more abundant in SCs than in EMs, although the difference was smaller. A higher expression of heat shock proteins (either 90, 70, or sHSP families) was also reported for SCs in a recent proteomic study [53]. The comparison was to immunocastrates (ICs), but they were vaccinated four weeks prior to slaughter and thus likely retained many EM metabolic features. There are many proteomic studies associating heat shock proteins (HSPs) with meat quality traits. Higher HSP (HSP27 and α-crystallin B) expression is associated with darker pork [39,54] or beef color [55]. With regard to meat tenderness, the results are more difficult to explain. The decreased abundance of HSP70 was shown for tender beef samples [51,56], and it was suggested that heat shock proteins provide resistance to oxidative stress and slow down the onset of muscle cellular death, which delays the rate of muscle aging and attenuates myofibrillar protein degradation (a process leading to muscle tenderization) [57,58]. These findings could explain the tougher meat of EMs. As reviewed by Lomiwes et al. [59], the chaperon activity of HSP70, involved in the active refolding of denatured proteins, is aided by sHSPs, having a complementary role in protecting proteins from denaturation and thus in delaying the apoptosis process. However, it is also important to note that HSP70 is ATP-dependent, while sHSPs are not [59], and the samples were taken 24 h post mortem (when ATP is depleted and thus only α-crystallin B remains active). Other proteomic studies, however, provide no straightforward conclusion with regard to sHSP expression and meat quality showing neither a negative [39,49] nor positive correlation [38] with meat tenderness and either no effect [39] or a positive correlation [41,60,61] with water holding capacity. 

In the case of troponin T, both fast skeletal and slow skeletal muscle isoforms were over-expressed in EMs. Troponin T has been recognized as a good marker of meat tenderness. Its breakdown and fragment appearance has been associated with either decreased shear force or increased sensorial tenderness [52]. In the present study, both troponin T spots have been identified as entire molecules (not fragments) in EMs, which would corroborate the higher meat toughness. However, it should be noted that in the case of troponin T degradation, polypeptide products ranging from 28 to 32 kDa are formed [37,62], which is relatively close to the entire molecule size. Because the methodology used in the present study does not enable us to distinguish small differences in molecular weight, the appearance of troponin T fragments (in accordance with numerous other fragments appearing) cannot be excluded.

Among the entire protein molecules that were overexpressed in EMs, several spots were identified as serum albumin (one of the main blood plasma proteins) and one spot as serotransferrin (a plasma protein involved in iron transfer). These two proteins are directly correlated with increased muscle carbonyl group content [63], which is in line with the results of the present study. Furthermore, a higher presence of plasma proteins could indicate a higher quantity of blood remains or a higher degree of vascularization. A denser muscle capillary network is observed in more oxidative muscles [64]. At the same time, a higher expression of malate dehydrogenase (an enzyme involved in mitochondrial respiration [65] and of Na/K-transporting ATPase subunit alpha-1 (more expressed in oxidative muscle fibers [66,67]) was observed in EMs than in SCs, which could be an indication of the higher oxidative metabolism of EMs. Higher oxidative metabolism was reported for bulls exhibiting a lower proportion of white myofibers [68,69] or lower glycolytic metabolism [70,71,72]. The available literature on that issue with respect to pigs is limited and indicates only a hypertrophy of myofibers in EMs [73] and a somewhat higher oxidative metabolism in ICs than in SCs [74]. 

## 5. Conclusions 

The investigation of muscle proteomic profiles showed some indications of more oxidative LL muscle metabolism in EMs than in SCs. The higher quantities of protein fragments in EMs indicated a greater extent of proteolytic degradation. However, this did not affect meat toughness, as EMs exhibited higher meat shear force values than did SCs. The relatively tougher meat in EMs than in SCs can thus be related to the lower intramuscular fat content and lower water holding capacity of EMs, the latter of which is likely due to the higher level of protein oxidation in EMs than in SCs. 

## Figures and Tables

**Figure 1 animals-09-00074-f001:**
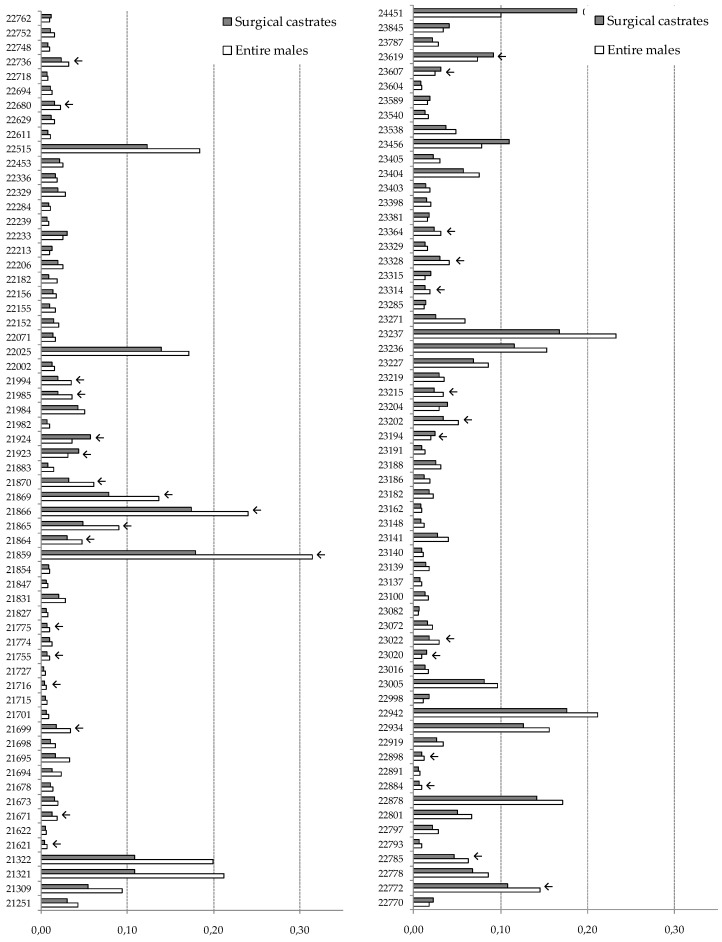
Relative abundancies (vol%) of the protein spots significantly differing (*p* < 0.05) between surgical castrates and entire males. The numbers corresponding to the relative abundancies represent the protein identification; the ones indicated by an arrow were identified by mass spectrometry.

**Figure 2 animals-09-00074-f002:**
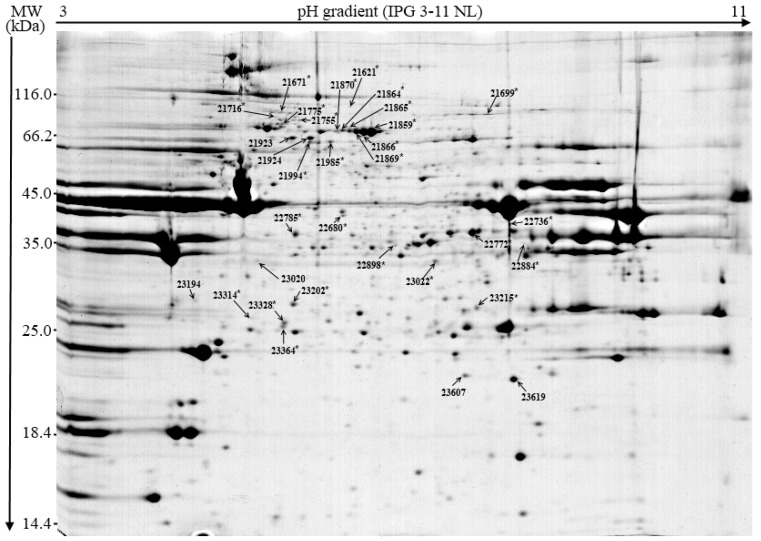
Two-dimensional electrophoresis gel image of the porcine longissimus lumborum muscle. The spots indicated by the arrows and identification (ID) number were identified by mass spectrometry; * denotes higher relative abundancy in samples from entire males.

**Table 1 animals-09-00074-t001:** Live weight and carcass traits of pigs included in the study according to sex group.

Live Weight and Carcass Traits	EMs	SCs	RMSE	*p*-Value
Live weight, kg	132.8	132.3	7.9	0.818
Carcass weight, kg	100.8	105.9	5.8	0.043
Back fat, mm	9.8	13.9	3.8	0.016
Muscle thickness, mm	71.2	81.1	6.9	0.002
Lean meat %	62.4	60.5	3.1	0.157

EMs: entire males; SCs: surgical castrates; RMSE: root-mean-square error. Back fat: minimal thickness of the fat on the top of the gluteus medius muscle. Muscle thickness: the shortest distance between the dorsal edge of the vertebral canal and the cranial end of the gluteus medius muscle.

**Table 2 animals-09-00074-t002:** Chemical composition of the longissimus dorsi, pars lumborum muscle according to sex group.

Chemical Traits	EMs	SCs	RMSE	*p*-Value
IMF, %	1.8	3.1	1.4	0.033
Proteins, %	24.1	23.6	0.6	0.118
Moisture, %	74.0	73.2	0.6	0.008
Collagen, mg/g	8.09	3.63	2.59	<0.001
Collagen solubility, %	22.8	11.5	8.8	0.005
Myoglobin, mg/g	1.26	1.45	0.28	0.118
Carbonyl, nmol/g protein	1.67	0.82	0.39	<0.001

IMF: intramuscular fat; EMs: entire males; SCs: surgical castrates; RMSE: root-mean-square error.

**Table 3 animals-09-00074-t003:** Meat quality traits of the longissimus dorsi, pars lumborum muscle according to sex group.

Meat Quality Traits	EMs	SCs	RMSE	*p*-Value
L*	55.4	52.7	2.4	0.011
a*	7.3	8.3	1.2	0.065
b*	3.4	2.1	1.0	0.006
Ultimate pH	5.35	5.37	0.08	0.417
Drip loss after 24 h, %	7.1	3.9	1.9	<0.001
Thawing loss, %	13.8	11.7	3.6	0.176
Cooking loss, %	34.1	28.8	3.7	0.002
Shear force, N	160.6	123.0	32.2	0.009

L*, a*, b* are color parameters; EMs: entire males; SCs: surgical castrates; RMSE: root-mean-square error.

**Table 4 animals-09-00074-t004:** List of protein spots identified by mass spectrometry.

ID	Consensus Protein Identity	UniProt ID ^a^	Mascot Score	% SC/MP ^b^	Theoretical Mr/Pi ^c^	Protein Integrity ^d^
Enzymes						
21621	Na/K-transporting ATPase subunit alpha-1	P05024	23	1/1	113920/5.36	Entire
21755	Na/K-transporting ATPase subunit alpha-1	P05024	18	1/1	113920/5.36	Fragment
22736	Beta enolase	Q1KYT0	152	8/3	47443/8.05	Fragment
22884	Creatine kinase M-type	Q5XLD3	151	8/3	43260/6.61	Fragment
23215	Creatine kinase M-type	Q5XLD3	177	8/3	43260/6.61	Fragment
22898	Malate dehydrogenase, cytoplasmic	P11708	93	6/2	36716/6.16	Entire
Blood plasma						
21699	Serotransferrin	P09571	122	6/3	78971/6.93	Entire
21859	Serum albumin	P08835	613	14/8	71643/6.08	Entire
21864	Serum albumin	P08835	226	6/3	71643/6.08	Entire
21865	Serum albumin	P08835	561	17/9	71643/6.08	Entire
21866	Serum albumin	P08835	605	14/8	71643/6.08	Entire
21869	Serum albumin	P08835	599	14/8	71643/6.08	Entire
21924	Serum albumin	P08835	61	3/2	71643/6.08	Entire
21923	Coagulation factor VIII	P12263	26	0/1	240467	Fragment
21985	Coagulation factor VIII	P12263	23	0/1	240467	Fragment
21994	Coagulation factor VIII	P12263	25	0/1	240467	Fragment
Chaperone						
21870	Heat shock 70kDa protein 6	Q04967	358	9/6	71522/5.77	Entire
23607	Alpha crystallin B chain	Q7M2W6	479	41/6	20116/6.76	Entire
23619	Alpha crystallin B chain	Q7M2W6	246	23/5	20116/6.76	Entire
Myofibrillar						
22772	Troponin T, fast skeletal muscle	Q75NG9	200	13/3	32157/6.05	Entire
23022	Troponin T, slow skeletal muscle	Q75ZZ6	146	12/3	31224/5,92	Entire
21671	Myosin heavy chain 2a	Q9TV63	104	1/2	223924/5.64	Fragment
21716	Myosin heavy chain 2a	Q9TV63	41	0/1	223924/5.64	Fragment
21755	Myosin heavy chain 2a	Q9TV63	114	1/2	223924/5.64	Fragment
22680	Actin alpha, skeletal muscle	P68137	295	15/5	42366/5.23	Fragment
22785	Actin alpha, skeletal muscle	P68137	378	15/4	42366/5.23	Fragment
23020	Actin alpha, skeletal muscle	P68137	199	10/3	42366/5.23	Fragment
23194	Actin alpha, skeletal muscle	P68137	169	8/3	42366/5.23	Fragment
23202	Actin alpha, skeletal muscle	P68137	281	10/3	42366/5.23	Fragment
23314	Actin alpha, skeletal muscle	P68137	160	10/3	42366/5.23	Fragment
23328	Actin alpha, skeletal muscle	P68137	276	10/3	42366/5.23	Fragment
23364	Actin alpha, skeletal muscle	P68137	279	10/3	42366/5.23	Fragment

^a^ Accession number derived from UniProt database (www.uniprot.org/uniprot); ^b^ Sequence coverage/number of matched peptides; ^c^ Theoretical molecular weight (MW)/isoelectric point (pI); ^d^ Integrity of the protein (either entire or fragment) based on the position of the spot on the gel in relation to the theoretical protein molecular weight.

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
