# Peer review of "Proteomic Profiles of the Longissimus Muscles of Entire Male and Castrated Pigs as Related to Meat Quality"

_animals, 2019, doi:10.3390/ani9030074_

Round 1

Reviewer 1 Report

Dear Authors,

I attached a pdf with some minor comments. However I have two main concerns which should be addressed before the paper can be accepted for publication:

1) 12 animals appear to be a very small sample size. Can the Author elaborate on the reasons why they chose this size? Do they expect to have a wider population sampled in the future? Is this a preliminary study? Do they believe the results could be extrapolated as they are? These aspects should be made very clear in the manuscript.

2) The differences in collagen content (regardless of collagen solubility) observed by the Authors between the two groups probably explain also the differences in shear force. I believe the Authors should consider this hypotesys and possibly include it in the manuscript.

Author Response

Dear Reviewer 1 ←

Thank you for your review. Below please find our answers.

I attached a pdf with some minor comments. However I have two main concerns which should be addressed before the paper can be accepted for publication:

1) 12 animals appear to be a very small sample size. Can the Author elaborate on the reasons why they chose this size? Do they expect to have a wider population sampled in the future? Is this a preliminary study? Do they believe the results could be extrapolated as they are? These aspects should be made very clear in the manuscript.

A: Proteomic analysis are time consuming, labour intensive and expensive, so 12 animals per treatment is not a small number (this makes in total 48 gels to analyse, i.e. 24 biological x 2 technical repetitions, plus adding the protein identification of selected spots). We are continuing with the research on the differences between EM and SC, but on additional aspects (muscle metabolism in regard to gene expression and histological traits) and in this regard, the current results will be useful.  

2) The differences in collagen content (regardless of collagen solubility) observed by the Authors between the two groups probably explain also the differences in shear force. I believe the Authors should consider this hypotesys and possibly include it in the manuscript.

A: The effect size in total collagen (EM vs. SC) was quite important, so the reviewer's comment is understandable; it would be the most obvious explanation for the differences in tenderness. However, there was no overall correlation between shear force and collagen. Moreover, correlation (significant) between shear force and collagen within group EM was even negative, so collagen content can’t explain increased meat toughness in EM.

The corrections, indicated by the reviewer directly in the manuscript, were addressed and corrected as follows:

Lines 11-12: corrected as suggested to »barrows (surgical castrates)« (now lines 11-12)

Line 35: corrected to »more common« (now line 35)

Line 41: »in EM« added (now line 42)

Line 51: according to journal instructions, sub-titles are only allowed in Results section. The paragraphs are not started with titles now (see lines 105, 137)

Lines 58 and 59: »DM fat« and »DM meat« deleted (now lines 58, 59).

Table 1: »DM fat« and »DM meat« were changed to »back fat« and »muscle thickness« (as in M&M). In addition, explanation was added to the table footnotes.

Table 2 and 3: LD was changed to »longissimus dorsi, pars lumborum« in accordance with the rest of the text.

Line 165 (+comment on line 171): In the figure 1 the identified spots were indicated by an arrow. Explanation was also added to the figure legend (now lines 171-172)

Table 4: Superscripts were corrected

Line 219: »also« added (now line 227)

Line 221: the text and references refer to colour parameters not to the myoglobin content. The text was changed accordingly (now lines 229-230 )

Line 231: »muscle« changed to »meat« (now line 240)

Lines 231-234: the sentences were rephrased: “Reduced water holding capacity and protein oxidation have been held responsible for increased meat toughness [35], which corroborates the results of the present study. Greater toughness of EM may be related to significant correlation between shear force and IMF (r=-0.57, P=0.003), between shear force and carbonyl groups content (r=0.51, P=0.010), whereas no significant correlations with shear force could be observed for either total collagen content (r=-0.07, P=0.752) or collagen solubility (r=-0.21, P=0.325).”  (now lines 239-244)

Line 274: the numbers refer to the number of fragments, as explained in the further sentence. The text was rephrased as follows: »…which could explain a 5-fold higher incidence of identified protein fragments (i.e. 15 vs. 3; Table 4) in EM than in SC…« (now lines 256-257)

Line 252: »androgen potential« changed to »androgens« (now line 261)

Line 252-253: please see our answer about the collagen above.

Line 254: »Muscle« changed to »muscles« (now line 264)

Line 254: Sentence rephrased to : »…may be degraded, it was concluded that they are not broken down to any …« (now lines 264-265).

Line 255: »this« added (now line 265)

Line 259: »fibre breaking strength« changed to »shear force« (now line 270)

Line 281: for this comment please see answer about the collagen above.

Best regards.

Reviewer 2 Report

The manuscript entitled, Proteomic profile of longissimus muscle of entire male and castrated pigs as related to meat quality, seeks to examine the effect not castrating boars has on meat quality when compared to castrated pigs. These effects were not only examined at the meat science level, but also at the molecular/biological level. This is a well executed study and the authors detect expected differences in meat quality. The molecular data indicated EM was tougher, had more collagen but it is less soluble, and 2D analysis indicated there may have been more proteolysis. The fact tat this was detected contradicts the the tenderness data and is where the authors probably spend too much of the document speculating about the data. The fact that the authors state more proteolysis occurred, but do not further explore products of proteolysis (such as troponion-T degradation) is very dangerous. Do the authors have any left over sample to conduct such analyses?

General Comment: The majority of the meat science and wet bench chemistry analyses refer to other papers. The authors should at least briefly describe the the methods.

Line 143: If the difference is not statistically significant, there is not a true difference to discuss.

Lines 147-148: IMF is presented as a relative change and moisture is presented as an absolute change. This should be consistent and the absolute change for moisture is 0.8%, not 1.1%.

Line 156: Tendencies were not declared in the methods section so inappropriate to discuss.

Lines 198-199: The way this reads, the reproduction tract weight 5 kg. The only difference between EM and SC was the testicles. Hard to believe the testicles weight 5 kg. This needs to be reconsidered.

Line 273: Apoptosis and postmortem proteolysis are two different processes catalyzed by different enzymes. This does not seem like a valid mechanism and need to be further explained.

Author Response

Dear Reviewer 2 ←

Thank you for your review. Below please find our answers.

The manuscript entitled, Proteomic profile of longissimus muscle of entire male and castrated pigs as related to meat quality, seeks to examine the effect not castrating boars has on meat quality when compared to castrated pigs. These effects were not only examined at the meat science level, but also at the molecular/biological level. This is a well executed study and the authors detect expected differences in meat quality. The molecular data indicated EM was tougher, had more collagen but it is less soluble, and 2D analysis indicated there may have been more proteolysis. The fact tat this was detected contradicts the the tenderness data and is where the authors probably spend too much of the document speculating about the data. The fact that the authors state more proteolysis occurred, but do not further explore products of proteolysis (such as troponion-T degradation) is very dangerous. Do the authors have any left over sample to conduct such analyses?

A: We have no left over of the sample. Moreover, the methodology that we used did not allow us to extend the analysis and to be more accurate with the conclusions about the protein fragmentation (as also explained in the discussion on troponin (see lines 299-303)

General Comment: The majority of the meat science and wet bench chemistry analyses refer to other papers. The authors should at least briefly describe the the methods.

A: Most of the meat and wet chemistry methodology refers to the classical well-known methods for measuring meat quality and composition published in numerous papers. It prolongs the paper and it is difficult to describe without repetition which can be problematic in view of auto-plagiarism. Proteomic part was detailed.

Line 143: If the difference is not statistically significant, there is not a true difference to discuss.

A: It is not discussed – this sentence appears in results. Lean meat content is directly derived (equation) i.e. is result of backfat and muscle thickness, that’s why the diction “resulting in” was used. (now line 145)

Lines 147-148: IMF is presented as a relative change and moisture is presented as an absolute change. This should be consistent and the absolute change for moisture is 0.8%, not 1.1%.

A: Not so - both differences (IMF and moisture) were presented as a relative change (in relation to SC).

Line 156: Tendencies were not declared in the methods section so inappropriate to discuss.

A: Tendencies are now declared in material and method (now line 141)

Lines 198-199: The way this reads, the reproduction tract weight 5 kg. The only difference between EM and SC was the testicles. Hard to believe the testicles weight 5 kg. This needs to be reconsidered.

A: »reproductive tract« as written in the text is not only the testicles, it denotes also accessory glands, and all the adjacent tissues (skin, connective tissue of the scrotal region…) that is removed; possible difference in the weight of intestines might have contributed as well to this difference in dressing percentage. Sentence rephrased (now lines 206-207).

Line 273: Apoptosis and postmortem proteolysis are two different processes catalyzed by different enzymes. This does not seem like a valid mechanism and need to be further explained.

A: We agree, but protein degradation begins after the cell death. We refer to previous studies presenting the connection between the two processes. The sentence was rephrased (see lines 282-285).

Best regards.

Reviewer 3 Report

Review of Proteomic profile of longissimus muscle of entire male and castrated pigs as related to meat quality by Škrlep et al.

The manuscript describes experiments comparing the proteome profiles of muscle tissue in relation to meat quality in both entire en castrated boars. In general the experiments were well performed, and well described. Some comments below, and modifications are required.

Comments and suggestions

Simple summary: Is it simple to mention the latin name of the muscle?

Line (L)13: males (add “s”)

L15: replace last with latter

L19: of THE same age

L55: is there a slaughter day effect, or were all Animals slaughtered at the same day?

L68: Because the samples for proteome analysis were taken after 24 hours there may be loss of fragmented proteins. How did the authors take this into consideration? This may affect the results of L102-103, and L132, and L174-175

L105: Is “Identification of protein spots” a header? This reviewer prefers to make such subheaders to improve readability. Idem: L136

L143: If it is not significant, it is not different (scientifically)

L151: Materials and Methods. Please move the text (measured as protein carbonyl groups content)

L175-176: The rest of the spots, where the observed difference was small, were considered 175 as entire protein molecules. Why? Do you have evidence for this statement?

In the Discussion section at several places: In agreement with our results. I recommend to reverse this statement: your results were in agreement with the previously reported results

L220: If it is not significant it should not be reported as a difference

L225: the sentence contains a number of English language errors: The majority.... showed....pointed out (past tense!)

L231-235: Long and difficult to read sentence. Please rewrite

L251: indicated

L263: form the?? from the??

L304-307: This is a rather disappointing conclusion from a well-performed study. Why? Please try to explain further!

L313: due to a higher

Author Response

Dear Reviewer 3 ←

Thank you for your review. Below please find our answers.

The manuscript describes experiments comparing the proteome profiles of muscle tissue in relation to meat quality in both entire en castrated boars. In general the experiments were well performed, and well described. Some comments below, and modifications are required.

Comments and suggestions

Simple summary: Is it simple to mention the latin name of the muscle?

A: Changed to »loin muscle« (now line 11)

Line (L)13: males (add “s”)

A: corrected (line 13).

L15: replace last with latter

A: Corrected (now line 15).

L19: of THE same age

A: Corrected (now line 19).

L55: is there a slaughter day effect, or were all Animals slaughtered at the same day?

A: all animals were slaughtered on the same day.

L68: Because the samples for proteome analysis were taken after 24 hours there may be loss of fragmented proteins. How did the authors take this into consideration? This may affect the results of L102-103, and L132, and L174-175

A: If you mean the loss of proteins with meat juice – there was none of it lost (for proteomic analysis we sampled immediately after cutting). If you mean potential loss due to degradation to fragments during first 24 h post mortem, we are aware that proteolytic process (in particular due to calpains) starts. However, it is the usual procedure for meat quality measurements and we aimed at testing proteomic profile in the same time frame.

L105: Is “Identification of protein spots” a header? This reviewer prefers to make such subheaders to improve readability. Idem: L136

A: The »subheaders« were removed because journal’s instructions do not foresee subtitles in M&M section, only in Results (now lines 137, 105)

L143: If it is not significant, it is not different (scientifically)

A: Indeed. However in this case, the lean meat percentage is directly derived from the two mentioned variables (backfat and muscle thickness) i.e. equation, that’s why the diction “which resulted in”. Sometimes, even if the effect is insignificant, it is not marginal. Difference of 2% in LMP, even if not significant, is not negligible.

L151: Materials and Methods. Please move the text (measured as protein carbonyl groups content)

A: This part was rewritten (now line 155)

L175-176: The rest of the spots, where the observed difference was small, were considered 175 as entire protein molecules. Why? Do you have evidence for this statement?

A: Methodology limitations. We can assume that the molecules are entire or fragments, based on the small difference between observed and theoretical MW. We tried to elucidate that (results, line 181) and discussion on troponin (lines 198-302).

In the Discussion section at several places: In agreement with our results. I recommend to reverse this statement: your results were in agreement with the previously reported results

A: text rewritten (see line 212).

L220: If it is not significant it should not be reported as a difference

A: This part referring to colour was rewritten (lines 227-230).

L225: the sentence contains a number of English language errors: The majority.... showed....pointed out (past tense!)

A: Corrected as suggested (now lines 233-234)

L231-235: Long and difficult to read sentence. Please rewrite

A: The sentence was rewritten (now lines 240-244).

L251: indicated

A: Corrected to “which is indicative of« (now line 261).

L263: form the?? from the??

A: Corrected (now line 273).

L304-307: This is a rather disappointing conclusion from a well-performed study. Why? Please try to explain further!

A: The sentence was awkward and unclear so it was omitted (lines 315-318), but we added to the conclusion, that there are some indications of more oxidative muscle metabolism in EM (see lines 320-321).

L313: due to a higher

A: Corrected (now line 325).

Best regards.
